# Evaluation of Bacterial Viability for Fecal Microbiota Transplantation: Impact of Thawing Temperature and Storage Time

**DOI:** 10.3390/microorganisms12071294

**Published:** 2024-06-25

**Authors:** Paolo Bottino, Daria Vay, Christian Leli, Lidia Ferrara, Valentina Pizzo, Franca Gotta, Alessio Raiteri, Fabio Rapallo, Annalisa Roveta, Antonio Maconi, Andrea Rocchetti

**Affiliations:** 1Microbiology and Virology Laboratory, Azienda Ospedaliera Universitaria “SS. Antonio e Biagio e C. Arrigo”, 15121 Alessandria, Italy; dvay@ospedale.al.it (D.V.); christian.leli@ospedale.al.it (C.L.); lidia.ferrara@ospedale.al.it (L.F.); valentina.pizzo@ospedale.al.it (V.P.); fgotta@ospedale.al.it (F.G.); arocchetti@ospedale.al.it (A.R.); 2Department of Science and Technological Innovation, University of Eastern Piedmont, 15121 Alessandria, Italy; alessioraiteri@gmail.com; 3Department of Economics, University of Genova, 16126 Genova, Italy; fabio.rapallo@unige.it; 4Research Training Innovation Infrastructure, Research and Innovation Department (DAIRI), Azienda Ospedaliera Universitaria “SS. Antonio e Biagio e C. Arrigo”, 15121 Alessandria, Italy; aroveta@ospedale.al.it (A.R.); amaconi@ospedale.al.it (A.M.)

**Keywords:** Fecal Microbiota Transplantation (FMT), stool bank, *Clostridiodes difficile* infection, bacterial viability

## Abstract

Fecal Microbiota Transplantation (FMT) represents a promising therapeutic tool under study for several purposes and is currently applied to the treatment of recurrent *Clostridioides difficile* infection. However, since the use of fresh stool was affected by several issues linked to donor screening, the development of a frozen stool bank is a reliable option to standardize FMT procedures. Nevertheless, different environmental factors impact microbial viability. Herein, we report the effect of different thawing temperatures and storage conditions on bacterial suspensions in the FMT procedure. In total, 20 stool samples were divided into aliquots and tested across a combination of different storing periods (15, 30; 90 days) and thawing procedures (4 °C overnight, room temperature for 1 h; 37 °C for 5 min). Focusing on storage time, our data showed a significant reduction in viability for aerobic and anaerobic bacteria after thawing for 15 days, while no further reductions were observed until after 90 days. Instead, among the different thawing procedures, no significant differences were observed for aerobic bacteria, while for anaerobes, thawing at 37 °C for 5 min was more effective in preserving the bacterial viability. In conclusion, the frozen fecal microbiota remained viable for at least three months, with an excellent recovery rate in all three thawing conditions.

## 1. Introduction

Fecal Microbiota Transplantation (FMT) involves infusing liquid filtrated feces from a healthy donor into the gut of a recipient for the treatment of specific diseases related to unbalanced intestinal microbiota composition, a condition known as dysbiosis [1]. This procedure represents a straightforward therapy for the treatment of recurrent *Clostridioides difficile* infection (rCDI) and it is still under investigation as a potential therapeutic approach for other diseases [2,3,4,5]. Moreover, it has proven to be safe, well tolerated and effective [2,6,7] with a high therapeutic rate (more than 90%) and significantly negligible rate of adverse effects [8]. For the FMT procedure, the donor’s feces are emulsioned and filtered in order to obtain a liquid suspension that contains a eubiotic and balanced bacterial composition. Subsequently, it is transferred via colonoscopy, nasogastric tube, enemas or oral administration (lyophilized product) to the recipient’s gut with the aim of reestablishing the dysbiotic microbiota [9,10]. The fecal matter used for the transplant should be administered as a fresh solution (provided within 6 h of collection from the donor) or stored at −80 °C for up to six months for multiple FMT procedures. The most significant limitation of FMT infusion is the availability of healthy donors suitable for specimen collection after screening procedures [11]. Indeed, for safety reasons, the donor’s evaluation involves an extensive test performed on blood and stool samples before the procedure begins (e.g., enteropathogenic and multidrug-resistance bacteria, enteric/hepatic/systemic viral infection and gastrointestinal parasitosis tests), thus resulting in a potential delay between FMT recipient’s enrolment and the administration of the infusion [12].

According to the above-mentioned arguments, the FMT approach based on frozen matter is pivotal in obtaining specimens ready for infusion when requested, thus mitigating delays related to donor selection and screening [13]. However, the standardization of the FMT procedure and the development of a biological repository (commonly referred to as a stool bank) still need optimization. Stool banks are centralized facilities that may operate at an institutional, national or international level and are currently established in different European countries [11,14]. Their role is the centralization of stool donor screening and the preparation of gut microbiota suspensions, allowing increased access to FMT procedures with a high level of safety, quality, and affordability.

The preparation and storing of Human Fecal Microbiota (HFM) for FMT purposes must be carried out following the indications of scientific societies in order to respect the requested standards of safety and traceability [1]. To ensure the preservation of viability bacteria in frozen specimens, the donor’s fecal samples need to be resuspended with saline solution and 10% glycerol in order to reduce frostbite-induced damage to bacteria. Subsequently, the emulsioned solution needs to be stored at −80 °C. After thawing, the sample should be used as soon as possible, without the possibility of refreezing [13]. In Italy, the regulatory agency for the FMT procedure is represented by the National Transplant Center, which provides all the requirements in a specific protocol [15]. Based on the Italian national guidelines, in order to improve FMT deployment, authorized centers are required to evaluate the efficacy of fecal microbiota before and after freeze–thawing procedures to improve FMT deployment.

The aim of this study was to investigate overall bacterial viability in response to different storage timespans (ranging from 15 to 90 days) and thawing conditions (4 °C, room temperature, 37 °C).

## 2. Materials and Methods

For this study, twenty fecal samples were collected from different healthy selected donors for the FMT procedure. Each specimen was collected in a sterile screw-capped container and processed within 6 h from collection. Briefly, 10 g of fresh feces was transferred in a sterile plastic bag with an internal filter (Seward BA6040/STR Stomacher^®^ 80 Biomaster Strainer Bag, Seward, Worthing, UK) and homogenized with 50 mL of sterile saline solution and 10% glycerol (Monico Spa, Venice, Italy) using the Stomacher 400 (Seward, UK) instrument for 60 s at low speed. Subsequently, the suspension was filtered throughout sterile gauze in order to remove the visible debris and collect bacterial cells in a sterile container. Finally, all samples were made up to a 50 mL volume with the addition of sterile saline solution and 10% glycerol and each of the twenty samples were divided into ten aliquots of 2 mL. All procedures were performed according to Italian guidelines for FMT in a BSL-2 hood [15].

Nine aliquots of bacterial suspensions derived from each sample were stored at −80 °C and tested across a combination of different storing periods (15 days: T1; 30 days: T2; 90 days: T3) and thawing procedures (4 °C overnight, room temperature for 1 h, 37 °C for 5 min). For each condition, 10 µL of diluted specimens (from 10^−4^ to 10^−7^) was seeded onto Blood agar, Schaedler agar (Biomèrieux, Marcy-l’Étoile, France) and incubated for 48 h at 37° C in aerobic and anaerobic conditions, respectively. Twenty aliquots, one from each specimen, not subjected to freezing were seeded as mentioned above and used as the baseline (T0).

The bacterial viability for each thawing/storing combination was evaluated in terms of average microbial load (in colony forming unit per ml, CFU/mL) and compared to the reference condition. Data were analyzed as Log_10_-transformed values using repeated measures analysis of variance (RMANOVA) based on two within-subject factors (time and treatment), F tests based on Wilks’ lambda for the significance of a factor and Fisher’s tests on transformed responses for contrasts. A *p*-value < 0.05 was considered significant. The statistical analysis was performed with R software (version 4.2.2) and SAS System (version 9.4).

## 3. Results

Immediately before freezing, the overall bacterial load was higher for anaerobic bacteria (9.37 ± 0.53 log_10_ CFU/mL) than for aerobic bacteria (8.55 ± 0.38 log_10_ CFU/mL). Table 1 and Figure 1 summarize the overall microbial loads for the tested combinations of thawing and storing conditions.

Focusing on aerobic bacteria, there was no significant difference between the three treatments (Wilks’ lambda = 0.8689, F = 1.36, *p*-value = 0.2824), while the factor time alone and combined with the treatment significantly affected bacterial viability (time, Wilks’ lambda = 0.3935, F = 8.73, *p*-value = 0.0010; Treatment × Time, Wilks’ lambda = 0.3326, F = 4.68, *p*-value = 0.0082) (Table 2).

Pairwise analysis performed on the time factor showed a significant reduction between T0 and T1 (*p*-value < 0.0001), while no further reductions in microbial load were observed for the subsequent time-points (T1 vs. T2, *p*-value = 0.0635; T2 vs. T3, *p*-value = 0.3064) (Table 2). However, the pairwise analysis performed for the factor treatment showed no significant differences between the tested conditions.

For anaerobic bacteria, the factors treatment and time showed a significant reduction in bacterial load both alone (treatment, Wilks’ lambda = 0.6236, F = 5.43, *p*-value = 0.0143; time, Wilks’ lambda = 0.6050, F = 3.70, *p*-value = 0.0324) and when combined (Treatment × Time, Wilks’ lambda = 0.1868, F = 10.16, *p*-value = 0.0002) (Table 3).

Pairwise analysis performed on factor time showed a significant reduction between T0 and T1 (*p*-value = 0.0026), while no further reductions in microbial load were observed for the subsequent time-points (T1 vs. T2, *p*-value = 0.3171; T2 vs. T3, *p*-value = 0.2516) (Table 3). Unlike aerobes, for anaerobic bacteria, the pairwise analysis on the factor treatment showed a significant difference between the thawing procedures performed at 4 °C overnight and 37 °C for 5 min (*p*-value = 0.0032). The latter condition provided better preservation of anaerobic bacteria viability.

## 4. Discussion

The FMT procedure has proven to ensure a high treatment rate with reduced side effects; however, due to technical, logistic and administrative issues, several healthcare settings are not still able to provide this treatment to patients who could benefit from it [16,17]. Moreover, the lack of defined regulatory procedures at the European and global level and the absence of strong evidence for the effectiveness of the FMT procedure in rCDI treatment represent underlying restrictions to its deployment [1]. For the aforementioned reasons, the preparation of frozen products could significantly improve the transplant process, allowing for high flexibility for both donors and recipients. Furthermore, the availability of a stool bank linked to authorized transplant centers could ensure an increase in FMT procedures, subsequently guaranteeing effective and safe treatment for patients affected by rCDI when needed [7,18]. The FDA recommends stool banks to improve the measurement of fecal microbiota viability as part of a long-term stability plan [19]. The guidelines suggest the following three possible methods to study the residual microbial viability of frozen products compared to fresh products: flow cytofluorimetry, qPCR and culture methods.

Flow cytometry employs fluorescent probes that target viable and/or dead cells and can be used to assess the remaining viable portion of intestinal microbiota in fecal samples after the freezing procedure. This method was shown to be a rapid, sensitive and quantitative assay. A four-fold reduction in the amount of live cells was observed after the freezing procedure was carried out. However, this technique requires standardized protocols and quality controls to ensure the reproducibility and comparability of results. Another limitation is the unknown fraction of cells that cannot be classified as alive or dead, which becomes dominant after freezing (57% of the total) [20].

The use of qPCR for the evaluation of 16S rDNA provided detailed information on the abundance of bacterial populations belonging to a microbial community. Viability assessment was carried out using propidium monoazide (PMA) to exclude dead cells from the following amplification process. This molecular technique is effective for semiquantitative purposes in simple synthetic communities but only allows qualitative assessments for richer and complex communities, such as the gut microbiota [21]. qPCR demonstrated a reduction in similarity between the fresh and frozen samples with longer storage periods; *Bacteroidetes* decreased in a time-dependent manner, while the abundance of Firmicutes was not significantly affected by storage for up to one year [22].

Finally, cultural methods of total viable colony forming units (CFUs) per ml of FMT product were performed with a series of dilutions on agar plates. Although this method does not allow the measurement of all the microorganisms that inhabit the human gut, it can provide an estimate of the overall viability rate, which is comparable across different storage times and FMT procedures [19]. In our study, which included twenty fecal samples stored for up to three months and thawed under different conditions, we observed an initial decrease in viable bacterial richness after 15 days, but relative stability at subsequent times (30 and 90 days). Better preservation was observed especially for anaerobes, which has also been reported in other studies [23,24]. Interestingly, bacterial viability was also observed to be stable for up to 12 months, both for aerobic and anaerobic bacteria [25].

However, in contrast to the results of the above-mentioned works, we showed that all the tested thawing conditions (4 °C, room temperature, 37 °C) allowed the recovery of a significant bacterial quota. Nevertheless, the latter condition (37 °C for five minutes) was the most effective in preserving anaerobic bacteria.

The main limitation of the present study was the lack of NGS data about the different bacterial taxa and how they could be affected by the freezing/thawing process mentioned above. For this reason, further in-depth studies based on 16s RNA analysis need to be performed. This would also help to overcome the limitations of cultural procedures and provide a complete overview of fecal microbiota and specific taxa at the genus/species level. Moreover, we did not evaluate the impact of the different tested conditions on FMT procedures applied to patients with rCDI, given that that long-term (−20 °C) preservation of transplanted feces can result in instability of the clinical outcome in FMT therapy [26].

Since the standardization of practical procedures is a critical issue for the centers involved in FMT procedures, selecting an appropriate method to evaluate the bacterial viability of a frozen product is a pivotal step. All the aforementioned advantages and limitations must be taken into account, in addition to the organization of the FMT center and the presence of a stool bank.

## 5. Conclusions

In conclusion, our data suggest that frozen fecal microbiota remains viable for at least three months, with an excellent recovery rate under all three thawing conditions (4 °C, room temperature, 37 °C), although the latter is more effective in preserving anaerobic bacteria. Moreover, in this study, we reported a cheaper procedure accessible to all clinical microbiology laboratories that can be routinely performed in FMT centers in order to evaluate bacterial viability.

## Figures and Tables

**Figure 1 microorganisms-12-01294-f001:**
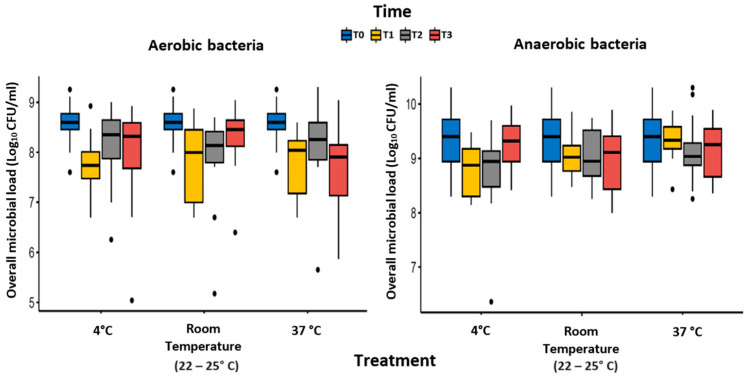
Analysis of overall microbial load for the different thawing and storing conditions for aerobic and anaerobic bacteria.

**Table 1 microorganisms-12-01294-t001:** Analysis of overall microbial load for the different thawing and storing conditions for aerobic and anaerobic bacteria.

Treatment	Time	Overall Microbial Load (Log_10_ CFU/mL)			
		Aerobic Bacteria		Anaerobic Bacteria	
		Mean	Standard Deviation	Mean	Standard Deviation
	T0	8.55	0.384	9.37	0.535
4 °C	T1	7.73	0.756	8.80	0.475
	T2	8.13	0.756	8.79	0.694
	T3	8.01	0.924	9.25	0.465
Room Temperature (22–25 °C)	T1	7.88	0.781	9.03	0.373
	T2	7.88	0.842	9.04	0.488
	T3	8.30	0.583	8.94	0.628
37 °C	T1	7.77	0.643	9.36	0.332
	T2	8.19	0.758	9.14	0.511
	T3	7.63	0.825	9.14	0.502

T0: 0 days; T1: 15 days; T2: 30 days; T3: 90 days.

**Table 2 microorganisms-12-01294-t002:** RMANOVA analysis and pairwise comparison for aerobic bacteria.

RMANOVA Analysis			
	Wilk’s Lambda	F Value	*p*-Value
Treatment	0.8689	1.36	0.2824
Time	0.3935	8.73	0.0010
Treatment × Time	0.3326	4.68	0.0082

Time: pairwise comparisons (*p*-value)			
	T0	T1	T2
T1	<0.0001		
T2	0.0040	0.0635	
T3	0.0014	0.1574	0.3064

T0: 0 days; T1: 15 days; T2: 30 days; T3: 90 days.

**Table 3 microorganisms-12-01294-t003:** RMANOVA analysis and pairwise comparison for anaerobic bacteria.

RMANOVA Analysis			
	Wilk’s Lambda	F Value	*p*-Value
Treatment	0.6237	5.43	0.0143
Time	0.6050	3.70	0.0324
Treatment × Time	0.1868	10.16	0.0002

Time: pairwise comparisons (*p*-value)			
	T0	T1	T2
T1	0.0026		
T2	0.0053	0.3171	
T3	0.0517	0.6690	0.2516

T0: 0 days; T1: 15 days; T2: 30 days; T3: 90 days.

## Data Availability

The data contained in this manuscript are available upon request.

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
