# Peer review of "Evaluation of Bacterial Viability for Fecal Microbiota Transplantation: Impact of Thawing Temperature and Storage Time"

_microorganisms, 2024, doi:10.3390/microorganisms12071294_

Round 1
Reviewer 1 Report
Comments and Suggestions for Authors
It is an interesting study regarding the impact of thawing conditions in bacterial viability of frozen fecal microbiota.
In the ms title the authors mention the use of FMT in rCDI.
However, within the text there is no corresponding correlation and reference to the role of FMT in the specific infection (CDI) which constitutes the only documented recommendation for the therapeutic application of FMT.
Moreover, the presence of specific bacterial taxa that affect the therapeutic benefit of FMT should also be assessed probably by using NGS.
Author Response
We are very grateful for the advices and the comments provided by the reviewers in order to improve our manuscript. We have taken in account all these corrections in the revised version of the manuscript. Below I have stated our response to each of the suggestions made by the reviewers. We hope that the modifications made in the revised manuscript and that the way we addressed the comments made by the reviewers meet the expectations. Otherwise, we are open to new improvements.
Sincerely
Paolo Bottino and co-authors
- In the ms title the authors mention the use of FMT in rCDI. However, within the text there is no corresponding correlation and reference to the role of FMT in the specific infection (CDI) which constitutes the only documented recommendation for the therapeutic application of FMT.
We thank the reviewer for this observation. Actually, in the present work there no data about correlation between FMT starting from frozen stool material and C. difficile infection. For this reason, we reformulated the title removing references to the FMT in rCDI. The new title “Evaluation of bacterial viability for fecal microbiota transplantation: impact of thawing temperature and storage time” is focused on data presented in the paper about only the impact of tested condition on bacterial viability. For the same reasons, we reformulated the abstract (lines 18-22). Nevertheless, references to use of FMT in rCDI were reported along the text (lines 40-41, 189-191 and references 7,16, 17).
- Moreover, the presence of specific bacterial taxa that affect the therapeutic benefit of FMT should also be assessed probably by using NGS.
We agree with the reviewer for the proposed observation. However, this represents the main limitation of the present work and better reported it at lines 227-232. We hope to could provides these data in a future work, to complete results herein reported. Moreover, a further limitation not previously mentioned was added together the abovementioned aspect (lines 232-235).
All the changes mentioned and reported in the text have been highlighted alongside the manuscript.
Reviewer 2 Report
Comments and Suggestions for Authors
To explore the effects of different thawing temperatures and storage conditions on bacterial suspensions applied to the FMT procedure, the authors divided 20 stool samples into aliquots and tested them under a combination of different storage periods (15 days, 30 days, 90 days) and thawing procedures (overnight at 4°C, 1 h at room temperature, and 5 min at 37°C). The results showed that the frozen fecal microbiota could survive for at least 3 months, with excellent recovery under all 3 thawing conditions. The structure of the article is complete, but the research content and the presentation of the results are too simple.
1. The title of the article is inappropriate. The study did not involve research on recurrent Clostridium difficile infection. Therefore, it is recommended to change the title.
2. How does the extensive testing of blood and stool samples from donors mentioned in the preface relate to this study?
3. On page 3, lines 81-83, what is the basis for investigating the effects of different thawing conditions (4°C, room temperature, 37°C) on overall bacterial viability?
4. On page 3, lines 85-91, in the experimental method, 20 samples were collected and divided into 10 aliquots, please explain whether each sample is divided into 10 aliquots, or the 20 samples are mixed into 10 aliquots?
5. Page 3, lines 96-102, have replicates been performed under the same thawing and storage conditions in this study?
6. Page 3, lines 99-105, Are the two solid plates described in the study method suitable for the culture of all fecal bacteria?
7. Flow cytometry and qPCR were discussed at length in the discussion, so why not focus on the experimental methods and results of this study?
Author Response
We are very grateful for the advices and the comments provided by the reviewers in order to improve our manuscript. We have taken in account all these corrections in the revised version of the manuscript. Below I have stated our response to each of the suggestions made by the reviewers. We hope that the modifications made in the revised manuscript and that the way we addressed the comments made by the reviewers meet the expectations. Otherwise, we are open to new improvements.
Sincerely
Paolo Bottino and co-authors
- The title of the article is inappropriate. The study did not involve research on recurrent Clostridium difficile infection. Therefore, it is recommended to change the title.
As suggested from both reviewers we reformulated the title as following: “Evaluation of bacterial viability for fecal microbiota transplantation: impact of thawing temperature and storage time”. In this way, we hope it could be appreciated since better focused on the impact of tested condition on bacterial viability. Confounding references to rCDI related to the FMT were removed. For the same reasons, we reformulated the abstract (lines 18-22).
- How does the extensive testing of blood and stool samples from donors mentioned in the preface relate to this study?
In our experience, the selection of suitable donors represents the main limitation to a wide application of FMT due the presence in several cases of at least one exclusion criteria. These aspects were more critical if considering to start from fresh stool samples. The approach based on frozen material was useful to overcome the lackness of suitable donors. For this reason, we included this aspect in our work. However, in order to not lengthening the preface, we removed the table 1, resuming some critical aspects at lines 54-55.
- On page 3, lines 81-83, what is the basis for investigating the effects of different thawing conditions (4°C, room temperature, 37°C) on overall bacterial viability?
For the testing condition we referred to another published (https://doi.org/10.1111/jam.14522) also mentioned in the national program for FMT, in order to respect the national guidelines:
Starting from the abovementioned work, we assessed the bacterial viability at the same time and using the mentioned temperature of thawing (37° C). Moreover, we added other two conditions in order to provide a better evaluation of others suitable options that could be used by the different laboratories also according their workflow or internal planning.
- On page 3, lines 85-91, in the experimental method, 20 samples were collected and divided into 10 aliquots, please explain whether each sample is divided into 10 aliquots, or the 20 samples are mixed into 10 aliquots?
In the present work we collected 20 samples and each one, after the preparation performed according to the national FMT program, was divided into 10 aliquots. No mixing of different samples was carried out. We reformulated the sentences in order to better specify this aspect (lines 90-91). Moreover, we added the reference (line 92) for the followed protocol, yet mentioned in the introduction (reference 15) with the link the PDF file that mentioned all technical aspects.
- Page 3, lines 96-102, have replicates been performed under the same thawing and storage conditions in this study?
For each thawing/storing condition were tested 20 samples derived from the respective collected specimens. Thus, a total of twenty replicates under same thawing and storage conditions were performed. Other 20 fresh samples (one aliquot for each one) were tested without the reported condition as baseline control. We modified the text, adding these details in the manuscript (line 94, 99-100).
- Page 3, lines 99-105, Are the two solid plates described in the study method suitable for the culture of all fecal bacteria?
We selected these agar media referring to the abovementioned work (https://doi.org/10.1111/jam.14522) reported in the Italian FMT program. The used agar plates were Trypticase Soy Agar with blood, Columbia CNA with agar and Schaedler Agar. However, in the paragraph of results no differentiation of bacterial load count were provided for the different used plates.
For this reason, we decided to select only the media Agar blood and Schaedler in order to allow the growth of a higher number of microorganisms, respectively aerobes and anaerobes, without the pressure of several selective agents in the agar plates that could affect the viability of some taxa.
- Flow cytometry and qPCR were discussed at length in the discussion, so why not focus on the experimental methods and results of this study?
We agree with the reviewer about the proposed suggestion. For this reason, we added some references (23-26) to improve the data herein reported, discussing them with other results (lines 220-226). We observed several similarities in terms of applied protocols and bacterial viability related to storing time, while no analysis was performed focusing on different thawing conditions. For this reason, we hope to could provide an added value with our results.
All the changes mentioned and reported in the text have been highlighted alongside the manuscript.
Round 2
Reviewer 2 Report
Comments and Suggestions for Authors
it may be accepted in present form